# Zonisamide Administration Improves Fatty Acid β-Oxidation in Parkinson’s Disease

**DOI:** 10.3390/cells8010014

**Published:** 2018-12-29

**Authors:** Shin-Ichi Ueno, Shinji Saiki, Motoki Fujimaki, Haruka Takeshige-Amano, Taku Hatano, Genko Oyama, Kei-Ichi Ishikawa, Akihiro Yamaguchi, Shuko Nojiri, Wado Akamatsu, Nobutaka Hattori

**Affiliations:** 1Department of Neurology, Juntendo University School of Medicine, Bunkyo-ku, Tokyo 113-8421, Japan; sueno@juntendo.ac.jp (S.-I.U.); mtfujima@juntendo.ac.jp (M.F.); h-amano@juntendo.ac.jp (H.T.-A.); thatano@juntendo.ac.jp (T.H.); g_oyama@juntendo.ac.jp (G.O.); kishikaw@juntendo.ac.jp (K.-I.I.); 2Center for Genomic and Regenerative Medicine, Juntendo University School of Medicine, Bunkyo-ku, Tokyo 113-8421, Japan; akihiro0781@gmail.com (A.Y.); awado@juntendo.ac.jp (W.A.); 3Medical Technology Innovation Center, Juntendo University, Bunkyo-ku, Tokyo 113-8421, Japan; s-nojiri@juntendo.ac.jp

**Keywords:** Parkinson’s disease, fatty acid β-oxidation, long-chain acylcarnitine

## Abstract

Although many experimental studies have shown the favorable effects of zonisamide on mitochondria using models of Parkinson’s disease (PD), the influence of zonisamide on metabolism in PD patients remains unclear. To assess metabolic status under zonisamide treatment in PD, we performed a pilot study using a comprehensive metabolome analysis. Plasma samples were collected for at least one year from 30 patients with PD: 10 without zonisamide medication and 20 with zonisamide medication. We performed comprehensive metabolome analyses of plasma with capillary electrophoresis time-of-flight mass spectrometry and liquid chromatography time-of-flight mass spectrometry. We also measured disease severity using Hoehn and Yahr (H&Y) staging and the Unified Parkinson’s Disease Rating Scale (UPDRS) motor section, and analyzed blood chemistry. In PD with zonisamide treatment, 15 long-chain acylcarnitines (LCACs) tended to be increased, of which four (AC(12:0), AC(12:1)-1, AC(16:1), and AC(16:2)) showed statistical significance. Of these, two LCACs (AC(16:1) and AC(16:2)) were also identified by partial least squares analysis. There was no association of any LCAC with age, disease severity, levodopa daily dose, or levodopa equivalent dose. Because an upregulation of LCACs implies improvement of mitochondrial β-oxidation, zonisamide might be beneficial for mitochondrial β-oxidation, which is suppressed in PD.

## 1. Introduction

Parkinson’s disease (PD) is the second most common neurodegenerative disorder and is characterized by motor symptoms such as tremor, rigidity, and akinesia [1]. Although symptomatic relief with levodopa medication and deep brain stimulation treatments are well established, there is currently no disease-modifying therapy based on disease pathogenesis [2]. In PD pathogenesis, mitochondrial dysfunction has been suggested to occur in the form of excessive oxidative stress, respiratory-chain insufficiency, and mitophagy flux abnormalities, among others [3]. However, therapeutic approaches targeted to these mitochondrial dysfunctions have not yet been used in the clinic.

Zonisamide (1,2-benzisoxazole-3-methanesulfonamide) is used clinically as an anti-epileptic agent and has been adapted to treat resting tremor as well as the wearing-off symptoms in PD patients in Japan [4]. Recently, zonisamide has become widely used, and its beneficial effects for the treatment of PD were shown to be mediated by several mechanisms including monoamine oxidase inhibition, channel blocking, and glutamate release inhibition [5]. The half-life of zonisamide is approximately 68 h and it is distributed to the whole body, including the brain and skeletal muscle, followed by its excretion in urine [6]. In cellular and animal models of PD, zonisamide was reported to have various neuroprotective effects via mitochondrial protection, including antioxidant effects through manganese superoxide dismutase (MnSOD) upregulation, adjustment of calcium influx, and brain derived neurotrophic factor (BDNF) signaling systems [7,8,9,10,11,12,13]. In addition, zonisamide was reported to reduce neuroinflammation through the modulation of microglia [14]. However, no human data associated with metabolic changes under zonisamide treatment have been reported in patients with PD.

Recent research conducted by our group and others has revealed that serum and/or plasma metabolomics are useful for revealing changes in metabolic pathways in PD [15,16,17,18,19]. Our group previously reported the analyses of capillary electrophoresis time-of-flight mass-spectrometry (CE-TOFMS) and liquid chromatography time-of-flight mass-spectrometry (LC-TOFMS), which showed decreased long-chain acylcarnitines (LCACs) and increased long-chain fatty acids (LCFAs) in PD patients, suggesting that β-oxidation insufficiency occurs primarily in early PD, and is not associated with levodopa medication [15].

In this context, to highlight the metabolic changes that occur with zonisamide administration, we conducted a case–case study, using well-established techniques, of 20 PD patients undergoing zonisamide treatment and 10 PD patients without zonisamide treatment.

## 2. Materials and Methods

### 2.1. Ethics Statement

This study protocol complied with the Declaration of Helsinki and was approved by the ethics committee of Juntendo University (2017135). Written informed consent was given by all participants.

### 2.2. Participants

PD was diagnosed according to the Movement Disorders Society Clinical Diagnostic Criteria for PD, with no dementia (Mini-Mental State Examination [MMSE] score ≥24) [20]. All participants were aged 51 to 82 years, were fluent in Japanese, and had no substantial neurological disease other than PD. Based on demographic and clinical characteristics obtained from medical records and directly from PD patients, the PD group comprised of 30 patients was subdivided into 10 not administered zonisamide and 20 with the oral administration of zonisamide (25 mg/day, after breakfast) (Table 1). Clinical symptoms were evaluated within a year from the beginning of the oral administration of zonisamide. Participants suffering from acute infectious diseases or acute/chronic renal or hepatic failure at the time of sample collection were excluded. No participant had a medical history or chronic illness of type 2 diabetes mellitus, skeletal muscle disease, cancer, aspiration pneumonia, inflammatory bowel disease, or collagen vascular disease.

### 2.3. Assessment of Clinical Symptoms

The clinical conditions of PD patients were evaluated using Hoehn and Yahr (H&Y) stages and the Unified Parkinson’s Disease Rating Scale (UPDRS)-III. For practical and ethical reasons, the H&Y stage and UPDRS-III rating were defined during the “on” state, when patients reported that the effect of the last dose of medication was optimal.

### 2.4. Blood Sample Collection

All fasting blood samples were collected at the outpatient department of Juntendo University Hospital between April 2015 and January 2018. Following an overnight fast (12–14 h), a plasma or serum sample was obtained using 7-mL EDTA-2Na blood collection tube (PN7R, Tokyo, SRL) or 8-mL blood collection tube(78447 SIM-L1008SQ3, Kyokuto Pharmaceutical Ind. Co. Ltd., Tokyo, Japan) followed by two or three inversions, respectively. The samples were then allowed to rest for 30–60 min at 4 °C, followed by centrifugation for 10 min at 2660× *g*. The plasma was then separated and placed in collection tubes that were stored in liquid nitrogen until analysis.

### 2.5. Metabolite Extraction

Metabolite extraction and metabolome analysis were conducted at Human Metabolome Technologies (HMT; Tsuruoka, Yamagata, Japan). For analysis with CE-TOFMS, 50 µL of plasma was added to 450 µL of methanol containing internal standards (Solution ID: H3304-1002, HMT) at 0 °C to inactivate enzymes. The extract solution was thoroughly mixed with 500 µL of chloroform and 200 µL of Milli-Q water and centrifuged at 2300× *g* at 4 °C for 5 min. A 400-µL sample of the upper aqueous layer was centrifugally filtered through a Millipore 5-kDa cutoff filter to remove proteins. The filtrate was centrifugally concentrated and resuspended in 50 µL of Milli-Q water for CE-TOFMS analysis at HMT. For analysis with LC-TOFMS, 300 µL of plasma was added to 900 µL of 1% formic acid/acetonitrile containing the internal standard solution (Solution ID: H3304-1002, HMT) at 0 °C (to inactivate enzymes). The solution was thoroughly mixed and centrifuged at 2300× *g* at 4 °C for 5 min. The supernatant was filtrated through a solid phase extraction column (Hybrid SPE phospholipid 55261-U, Supelco, Bellefonte, PA, USA) to remove phospholipids. The filtrate was desiccated and then dissolved with 120 µL of 50% isopropanol/Milli-Q for LC-TOFMS analysis at HMT.

### 2.6. Biochemical Measurements

Total serum creatine kinase was measured using automated enzymatic techniques (Sysmex Inc., Kobe, Japan). Serum creatinine was measured by an enzymatic method (KAINOS Laboratories, Inc., Tokyo, Japan). Serum aldolase was measured by a coupled enzyme assay (Alfresa Pharma, UK), while hemoglobin (Hb) A1c was measured by boronate-affinity high-performance liquid chromatography, in accordance with standard protocols.

### 2.7. Data Analysis

Peaks were extracted using MasterHands automatic integration software (Keio University, Tsuruoka, Yamagata, Japan) to obtain peak information including the *m*/*z* and peak area, as well as migration time (MT) for CE-TOFMS and retention time (RT) for LC-TOFMS. Signal peaks corresponding to isotopomers, adduct ions, and other product ions of known metabolites were excluded, and the remaining peaks were annotated according to the HMT metabolite database based on their *m*/*z* values with the MTs and RTs. To obtain the relative levels of each metabolite, areas of the annotated peaks were then normalized based on internal standard levels and sample volumes. For multivariate statistical analysis, partial least squares (PLS) analysis was performed using R [21].

### 2.8. Statistical Analysis

When a value was below the limit of detection, we assigned it half the minimum value of its compound. Wilcoxon tests were used to compare all individual analyses between PD patients with or without zonisamide. A *p*-value of less than 0.05 was considered statistically significant.

## 3. Results

### 3.1. Participants

The characteristics of PD patients included in the study are shown in Table 1. PD patients taking zonisamide had a significantly higher UPDRS-III tremor score at pretreatment of zonisamide compared with PD patients not taking zonisamide. There were no significant differences in age, sex, H&Y stage, disease duration, levodopa equivalent dose (LED), or levodopa daily dose (LDD) between PD patients with or without zonisamide treatment [22]. There was a statistically significant difference in UPDRS-III tremor score between the two groups (*p* = 0.00280).

### 3.2. Metabolomic Datasets

We analyzed the metabolomic profiles of blood plasma from 30 PD patients using CE-TOFMS and LC-TOFMS. Based on their *m*/*z* values, MTs and RTs, 383 metabolites were detected. Of these, 266 metabolites were detected in >50% of PD patients taking zonisamide and were analyzed in detail. As shown in Table 2, 12 metabolites were significantly changed in PD patients with zonisamide treatment compared with those without zonisamide (Appendix A). Four of these 12 were LCACs, and these were all upregulated. Next, we performed PLS analysis to identify which metabolites distinguished PD with zonisamide treatment from PD without zonisamide treatment (Figure 1). The top 10 metabolites, including three LCACs, are shown in Appendix A.

### 3.3. Increase of Long-Chain Acylcarnitine Levels in PD Patients with Zonisamide Treatment

Because four LCACs (AC(12:0), AC(12:1)-1, AC(16:1), and AC(16:2)), were significantly increased in PD patients with zonisamide treatment (Table 2), we tried to characterize the profile of LCACs detected, regardless of the lower limits of detection, for further investigation. Interestingly, 15 of 20 LCACs showed a trend toward an increase in PD patients with zonisamide treatment (Appendix A). Of these, seven LCACs (AC(12:0), AC(12:1)-1, AC(12:1)-2, AC(12:1)-3, AC(13:1)-1, AC(16:1), and AC(16:2)) were significantly increased (Appendix A). According to the PLS analysis, there were two LCACs (AC(16:1) and AC(16:2)) in the top 10 factors for differentiating between PD with zonisamide treatment and PD without zonisamide treatment. There were no significant correlations between any LCACs and zonisamide treatment duration or dosage (Appendix A). These results suggest that the increase in LCACs is not affected by zonisamide treatment period and dosage.

### 3.4. Association of LCACs with Age, H and Y Stage, UPDRS-III Scores, and LED

Plasma levels of LCACs are primarily decreased in the early stages of PD [15]. As shown in Table 1, there were no significant differences in patient characteristics between the two groups except for the UPDRS-III tremor score (Table 1). Thus, to precisely exclude the effects of disease severity and medication on the levels of LCACs, we performed correlation analyses. There were no significant correlations between any LCACs with age at sampling, H&Y stage, UPDRS-III scores, and LED (Table 3). In addition, there was a significant correlation between AC(16:2) and the amelioration of UPDRS-III tremor scores in PD with zonisamide that was efficacious (Appendix A). These results suggest that zonisamide, and not disease severity, aging, or LED, lead to the increase of LCACs.

### 3.5. Association of Skeletal Muscle Mass with Metabolites Associated with Fatty Acid β-Oxidation

Plasma LCACs are excreted mainly from the cellular mitochondria of skeletal muscles [23]. To exclude any effects of decreased skeletal muscle mass on fatty acid (FA) β-oxidation, we assessed the serum levels of creatine kinase, aldolase, and creatinine, and total blood levels of HbA1c (Appendix A). Although levels of aldolase, which is produced by skeletal muscles as well as the liver, were significantly downregulated by zonisamide, the levels of creatine kinase and creatinine, which correlate with the amount of skeletal muscle, did not change significantly. There was also no significant difference in HbA1c levels, which may influence FA β-oxidation activity. From these results, we concluded that zonisamide had almost no harmful effects on skeletal muscle, although we should consider the possibility that zonisamide might influence skeletal muscle.

### 3.6. Other Metabolites Significantly Changed by Zonisamide Treatment

As shown in Table 2, eight chemicals (except for four LCACs) were significantly changed in PD with zonisamide treatment. 1-methylnicotinamide, a metabolite of nicotinamide, and oleoylethanolamide, were associated with neuroprotective pathways [24]. In addition, the ratio of putrescine to ornithine was decreased in sporadic PD [25]. Although zonisamide treatment changed the levels of these three chemicals, the detailed metabolic pathways involved remain unclear because only one metabolite changed.

## 4. Discussion

This comprehensive metabolome analysis identified 12 metabolites that underwent significant changes with zonisamide treatment in PD. Four of the twelve metabolites were LCACs: (AC(12:0), AC(12:1)-1, AC(16:1), and AC(16:2)) and the PLS analysis using whole metabolome data identified two overlapping LCACs: AC(16:1) and AC(16:2). Furthermore, 15 LCACs had a tendency to be increased in PD patients with zonisamide, and only AC(16:2) was correlated to the amelioration of UPDRS-III tremor scores in PD with efficacious zonisamide. The LCAC levels did not correlate with disease severity or medication, suggesting that zonisamide itself may upregulate FA β-oxidation in patients with PD. Moreover, LCAC levels might define therapeutic reactivity with treatment of zonisamide.

Previously, we reported that decreased levels of LCACs were a useful biomarker for the diagnosis of PD in a double-cohort study [15]. β-oxidation of FAs occurs in mitochondria and peroxisomes; β-oxidation of short-to-long-chain FAs occurs in mitochondria, whereas β-oxidation of LCFAs occurs in peroxisomes. FAs with chain lengths of 14 or more carbons, which are the majority of those obtained from the diet or released from adipose tissues, need to undergo enzymatic reactions by the carnitine shuttle to enter mitochondria for β-oxidation [26]. In peroxisomes, acyl-coenzyme A may be incompletely oxidized, and medium-chain acyl-CoA produced in peroxisomes is transported to mitochondria to be oxidized. Because it is difficult for acyl-CoA to penetrate the outer membrane of mitochondria, it is transported by carnitine-acylcarnitine translocase after its conversion to ACs by carnitine palmitoyltransferase 1 (CPT1). LCACs are formed by CPT1 from acyl-CoA and carnitine to pass through the mitochondrial outer membrane. Zonisamide may improve β-oxidation via a mitochondrial protective effect [7,15].

Patients with PD can be subdivided into two types: tremor-dominant or postural instability/gait difficulty variants [27]. Our previous report revealed no association of disease severity as assessed by UPDRS-III with the levels of LCACs in PD [15]. The current study suggests that severe tremor, often observed in PD and treated with zonisamide, might affect the elevation of LCACs. Whilst aerobic exercise was reported to decrease levels of LCACs in obese patients, controls do not show changes in LCAC levels following short- or long-term exercise [28,29]. In the current cohort, the body mass index in PD without zonisamide was approximately 22, and there was no statistical difference between the two groups. In addition, blood samples in our study were obtained without exercise loading and with overnight fasting. Thus, we concluded that we did not need to consider more severe tremors in PD patients without zonisamide treatment.

Oxidative damage is a major contributor to cellular damage, which can result in neurodegenerative diseases [30]. Oxidative stress generates reactive oxygen species (ROS), which inflict oxidative damage on macromolecules such as lipids, DNA, and proteins. The imbalance between oxidative stress via ROS production and antioxidant factors in the body has an important role in PD development [31]. Mitochondrial ROS metabolism is catalyzed by mitochondrial antioxidant enzymes including MnSOD. Catabolic reactions via β-oxidation are also a major component of the regulation of ROS generation in mitochondria [32]. Taken together, in addition to the upregulation of MnSOD expression [7], an improvement in β-oxidation by zonisamide may protect against oxidative stress.

This study had some limitations. First, the interpretation of results requires caution because the sample size was small. Thus, it was difficult to conduct multivariate analysis to exclude confounding factors including age, sex, disease duration, and medication except for anti-parkinsonian drug. Second, there was an effect of other antiparkinsonian medications on zonisamide metabolism. A previous study demonstrated that levodopa/benserazide treatment elicited metabolic changes in skeletal muscle with a switch from lipid to carbohydrate metabolism, which may affect β-oxidation [33]. However, in the current cohort we did not detect any direct influences of LDD or LED on LCAC levels. In the future, de novo studies with additional treatment of zonisamide and including larger study populations should be performed, focusing on FA β-oxidation changes.

## 5. Conclusions

This is the first report showing the in vivo effects of zonisamide on human metabolism. We identified 12 metabolites, including four LCACs, whose increased levels were associated with an improvement in FA β-oxidation. Considering previous experimental studies of zonisamide treatment, it may have beneficial effects on mitochondrial function in patients with PD. Furthermore, zonisamide might affect the efficiency of the brain network to relieve the symptomatology and progression of PD.

## Figures and Tables

**Figure 1 cells-08-00014-f001:**
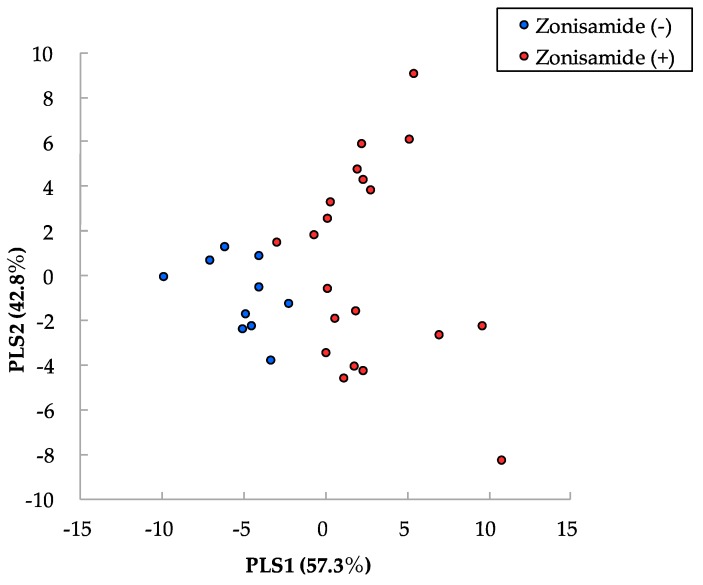
Partial least squares analysis (PLS) of observed metabolic profiles. Red indicates metabolites of PD with zonisamide, while blue indicates PD without zonisamide. PLS 1 indicates PD patients with or without zonisamide in PD, while PLS 2 indicates susceptibility of zonisamide in PD. PD: Parkinson’s disease, PLS: partial least analysis.

**Table 1 cells-08-00014-t001:** Demographics and clinical data of Parkinson’s disease with or without zonisamide.

Participants Characteristics, Mean (SD)	Zonisamide (−)	Zonisamide (+)	*p*-Value
**Number**	10	20	-
**Gender, female/male**	5/5	13/7	0.431 ^a^
**Age**	69.5 (6.7)	68.2 (1.9)	0.982
**Zonisamide**	-	38.7 (12.7)	-
**Treatment period by evaluation**	-	8.15 (3.75)	-
**Body mass index**	22.0 (3.4)	22.4 (0.75)	0.597
**H&Y**	2.1 (0.9)	2.3 (1.1)	0.762
**H&Y, each case number**	I(3), II(4), III(2), IV(1), V(0)	I(5), II(9), III(2), IV(3), V(1)	-
**Disease duration**	11.4 (6.9)	7.7 (4.0)	0.138
**UPDRS-III (pre-treatment)**	17.9 (12.4)	14.1 (3.1)	0.425
**UPDRS-III (post-treatment)**	-	13.6 (11.4)	-
**UPDRS-III-tremor (pre-treatment)**	0.6 (1.0)	2.6 (1.9)	0.00280
**UPDRS-III-tremor (post-treatment)**	-	2.2 (1.9)	-
**LED**	855.3 (305)	680.9 (458)	0.165
**LDD**	480 (209)	430 (280)	0.492

Abbreviations: SD: standard deviation, H&Y: Hoehn and Yahr stage, UPDRS: Unified Parkinson’s Disease Rating Scale, LED: levodopa equivalent dose, LDD: levodopa dose. ^a^: *p*-Value obtained by χ-squared test, other *p*-Values were obtained by Wilcoxon test.

**Table 2 cells-08-00014-t002:** Metabolites significantly changed in Parkinson’s disease differentiate between patients with or without zonisamide treatment.

Compound, Relative Area	Ratio of Zonisamide (+) to Zonisamide (−)	*p*-Value
**1-methylnicotinamide**	1.61	0.0294
**AC(12:0)**	1.73	0.0407
**AC(12:1)-1**	1.92	0.0405
**AC(16:1)**	1.75	0.0366
**AC(16:2)**	1.95	0.0054
**Glycerol**	0.843	0.0329
**Imidazolelactic acid**	1.50	0.0040
**Nervonic acid**	1.21	0.0068
**Oleoyl ethanolamine**	0.785	0.00146
**Ornithine**	1.32	0.0263
***S*-methylcysteine**	0.726	0.0294
**Succinic acid**	1.46	0.0199

AC: acylcarnitine. *p*-Value obtained by Wilcoxon test.

**Table 3 cells-08-00014-t003:** Association of LCACs and clinical parameters in all participants.

	Age		H&Y		UPDRS-III		LED	
	*r*	*p*-Value	*r*	*p*-Value	*r*	*p*-Value	*r*	*p*-Value
**AC(12:0)**	−0.293	0.115	0.128	0.499	0.240	0.200	0.287	0.123
**AC(12:1)−1**	−0.115	0.541	0.179	0.343	0.141	0.457	0.272	0.145
**AC(16:1)**	−0.199	0.291	0.0476	0.802	0.124	0.510	0.132	0.485
**AC(16:2)**	−0.110	0.562	0.185	0.326	0.153	0.417	0.0969	0.610

LCACs: long-chain acylcarnitines, H&Y: Hoehn and Yahr stage, UPDRS: Unified Parkinson’s Disease Rating Scale, LED: levodopa equivalent dose. *p*-Value obtained by Spearman’s rank correlation coefficient.

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
