# Peer review of "Zonisamide Administration Improves Fatty Acid β-Oxidation in Parkinson’s Disease"

_cells, 2018, doi:10.3390/cells8010014_

Round 1

Reviewer 1 Report

1. Introduction regarding the use of zonisamide to treat PD patients should be improved/extended.

2. Authors consider their study a “pilot study” (line 17); sample size is indeed “very small” and should be increased.

3. Line 58: “we conducted a case–control study” But no healthy controls (HCs) were included. Including HCs would increase the value of the study, e.g. does zonisamide bring the metabolite levels to control values?

4. Differences in UPDRS-III tremor scores between the two PSD groups was measured (between the group that did and the group that did not receive zonisamide) and they found significant differences. The authors discuss this point but do they really expect that this has no consequences for the presence/absence of specific metabolites in the two groups?

5. Did the treatment with zonisamide affect tremor or other PD symptoms in the treated PD patients?

6. In Results (line 158): seven LCACs (AC(12:0), AC(12:1)-1, AC(12:1)-2, AC(12:1)-3, AC(13:1)-1, AC(16:1), and AC(16:2)) were significantly increased (Table 2). Should be: Supplementary Table 2; in Table 2 only 4 LCACs are shown.

7. Why are not all seven significantly increased LCACs mentioned in Table 2?

8. Levels of aldolase were significantly downregulated by zonisamide; still the conclusion was that “there were no harmful effects of zonisamide on skeletal muscles” (line 182). Justified claim?

9. Lines 156-158: “15 of 20 LCACs showed a trend toward an increase in PD with zonisamide treatment (Supplementary Table 2).”  Line 187: “15 LCACs showed a tendency to be increased in PD  with zonisamide.” What is the definition here of “a trend” ? p < 0.3? Realistic?

10. Only four of the identified 12 metabolites with significant changes evoked by zonisamide treatment were discussed. Why? What about the other eight?

Author Response

Please find the PDF file.

Reviewer 2 Report

The authors challenge to present Zonisamide as a potential modulator of PD metabolism. In fact, they found that Zonisamide was able to modulate LCACs, demonstrating that this effect was Zonisamide exclusive and not correlated with the severity/stage of PD, LDD, or LED. Therefore, the authors have concluded and proposed that Zonisamide might be a beneficial strategy to tackle mitochondrial β-oxidation, which is suppressed in the majority of the cases of PD.   

The work is concise, well written and appropriate. However, in lines 45-48 the authors describe Zonisamide as potential neuroprotective PD drug. Notwithstanding, studies have also been suggesting Zonisamide as a potential disease-modifying strategy due to its actions in the modulation of microglial and astrocytic activities, which are important players on PD. Thus, this should also be considered as well. 

Line 70: Although the authors recognize that the N used in this work is to small to take robust interpretations, I would like to ask what was the statistical power used for the calculated number of patients? A precise description of the parameters used should be provided.

Line 72: The authors described that patients were evaluated within a year from beginning of the oral administration of Zonisamide. However, in which form were these evaluations performed? Daily? Monthly? This should be explained.

Line 138: 383 metabolites were identified, and the authors found that 266 of those were detected in more than 50% of the patients taking Zonisamide. Although it is an interesting result, a complete and discriminative analysis is missing. I would suggest the authors to draw a Venn diagram presenting the number of metabolites that are common to the experimental groups, and the numbers that are just restricted to each group, explaining what could be the principle for such differences.

Line168: It was assumed that Zonisamide per se was the responsible to the increase of LCACs, not being such increase correlated with the severity of the disease, or with age or even with Levodopa regimen. However, do the authors observe some (positive) correlation with this increase of LCACs promoted by Zonisamide and motor (non-motor) amelioration of PD patients? 

Line 222: What do the authors want to say with confounding factors?

Finally, do the authors have some indication if these observed results correlate with topological alterations/organization of the PD brain functional network after Zonisamide administration, and if its application may also affect the efficiency of the brain network to effectively relieve PD symptomatology and its progression?

Author Response

Please find the PDF file.

Reviewer 3 Report

Ueno et al examine the metabolome profile in a pilot study in PD patients which have been treated with zonisamide, a drug that is used to treat resting tremor as well as wearing off symptoms in PD patients.

The study is performed methodologically and technically well: the capillary electrophoresis time-of-flight mass-spectrometry and liquid chromatography time-of-flight mass spectrometry are state-of-the-art methods for such analyzes. The patient population has relatively more tremor-dominant patients, but overall, they are more likely to be treated with zonisamide in clinical practise. Otherwise, the patient populations between zonisamide treatment and controls are comparable. Core results are 4 increased levels of long-chain acylcarnitines (LCACs) (AC (12: 0), AC (12: 1) -1, AC (16: 1), and AC (16: 2)) in zonisamide-treated patients.

Major points: 

1. The individual duration of zonisamide treatment has to be shown in the baseline data.

2. The influence of zonisamide treatment duration and treatment dosage on LCACs levels should be examined.

Author Response

Please find the PDF file.

Round 2

Reviewer 1 Report

I now reviewed the revised version of manuscript cells-407645 ("Zonisamide administration improves fatty acid β-oxidation in Parkinson’s disease."). The manuscript has been significantly improved, but one important comment (Comment R1-2: "sample size is indeed “very small” and should be increased.") has been addressed by the authors by just stating: "We will try to increase the number of participants in a future study to confirm our results."  I would like to suggest that in case the other reviewer(s) do(es) not consider this as a major point (sample size - number of participants), as far as I am concerned the manuscript can then be accepted for publication in Cells.